

# Interactive effects between soil properties and bacterial communities in tomato rhizosphere under the application of microbial agents

Longxue Wei[1], Dongbo Zhao[1], Lianghai Guo[1], Jianjun Guo[1], Jinying Zhu[1], Yanting Pei[1], Peiyan Guan[2], Zhihui Guo[1], Huini Cui[1], Jiansheng Gao[1], Yongjun Li[1], Liang Zhang[1], Fuyan Wang[1] and Peng Liu[1]

[1] Dezhou Institute of Agricultural Science, Dezhou, Shandong, China
[2] Dezhou College, Dezhou, Shandong, China

Corresponding author
Peng Liu, wheatpb@163.com

## ABSTRACT

**Background:** The purpose of this study was to determine the impact of microbial agents on physicochemical properties, enzyme activities, and bacterial communities in tomato (*Solanum lycopersicum* L.) rhizosphere soil in order to provide a theoretical foundation for the stability of tomato rhizosphere soil microecology and the ecological preservation of farmland soil.
**Methods:** Tomato rhizosphere soils treated with various microbial agents, as well as an untreated control (FQ_CK), were analyzed. The effects of different microbial agents on the physicochemical properties, enzyme activities, and bacterial community structure of tomato rhizosphere soil after 2 years of continuous cropping were analyzed using 16S rRNA high-throughput sequencing technique.
**Results:** With the application of microbial agents, microbial agent treatment (*Bacillus subtilis* (FQ_T1), *Trichoderma harzianum* (FQ_T2), *Bacillus amyloliquefaciens* (FQ_T3), *Verticillium thuringiensis* (FQ_T4), and *Verticillium purpureum* (FQ_T5)) was beneficial for improving the physicochemical properties and enzyme activity of tomato rhizosphere soil after 2 years of continuous cropping. Compared with the control FQ_CK, bacterial treatment increased the richness of bacterial communities, total number of species, and overall relative abundance of beneficial bacterial phylum and genus, to a certain extent. It had a positive impact on microbial structure and function, improved the interaction and stability of species ecological networks, and was conducive to the stability and sustainability of tomato rhizosphere soil microbial ecosystem functions.

# INTRODUCTION

The tomato (*Solanum lycopersicum* L.) stands as one of the most significant vegetable crops cultivated globally (*Singh & Siddiqui, 2015*). As intensive production increases and facility agriculture develops, continuous cropping obstacles such as serious soil-borne diseases have become increasingly prominent and result in reduced quality and yield,

which seriously restrict the sustainable development of the tomato industry (*Su et al., 2023*). Soil is the carrier of plant production and also a fundamental element of the ecological environment. Soil microorganisms act as 'regulators' of soil nutrient cycling in the process of nutrient transformation, controlling the direction of soil nutrient cycling, the types of nutrient element compounds, and exchange fluxes (*Adomako, Roiloa & Yu, 2022*; *Luo et al., 2019*). Soil microbial communities and diversity regulate the multifunctionality of soil ecosystems, thereby influencing the soil's resistance and resilience, which are essential for maintaining soil fertility and sustainable production (*Hemkemeyer et al., 2021*). Understanding the ecological processes involved in microbial community assembly helps to determine how microbial community composition responds to environmental changes (*Coban, De Deyn & van der Ploeg, 2022*; *Xu et al., 2015*). Continuous cropping can disrupt the microbial community structure in the rhizosphere, impair the survival of beneficial organisms, foster the growth of pathogenic bacteria, intensify the occurrence of soil-borne diseases, and subsequently impact crop growth, resulting in yield reduction (*Feng et al., 2023*; *Xiao et al., 2024*). Studies have found that rhizosphere microorganisms are crucial in the onset and control of soil-borne diseases. Plants can recruit beneficial microorganisms in the rhizosphere *via* root exudates, thereby enhancing disease resistance through the antagonism, competition, and induction of systemic resistance by these beneficial microorganisms (*Su et al., 2023*). Research by *Fu et al. (2017)* indicates that soil-related issues stemming from continuous monoculture have hindered the sustainable development of China's tomato industry. Tomato yield was significantly positively correlated with soil available potassium (AK) and soil microbial community functional diversity index.

Microbial inoculants are those based on one or more functional or bacterial strains applied to the soil as a substitute for traditional inorganic fertilizers (bio-fertilizers), which can be used for bioremediation (*Yang et al., 2023*), and enhancement of soil properties (*Jiang et al., 2022*). Many scholars have learned about *Bacillus amyloliquefaciens*, strain FZB42 RhizoVital 42® (RV) (*Bouček et al., 2023*), *Bacillus subtilis* (*Kamaraj, John & Annaiyan, 2022*), *Trichoderma harzianum* (*Shalaby et al., 2022*), and *Bacillus amyloliquefaciens* (*Liu et al., 2018*) through inoculation. The results show that it has a certain effect on improving tomato fruit quality, reducing tomato root knot nematodes, reducing seedling wilt disease, and inhibiting tomato disease occurrence. A more accurate understanding of the ecology and mode of action of inoculant strains is key to optimizing their efficacy and guiding their targeted use to address key constraints to crop production (*Gu et al., 2024a*).

At present, it is very important to predict the role of rhizosphere microorganisms in regulating the function of plant-microorganism ecosystems. This study used 16S rRNA high-throughput sequencing technique to investigate the effects of different microbial preparations on the physicochemical properties and bacterial community composition of tomato rhizosphere soil after 2 years of continuous cultivation. Firstly, this study revealed the effects of different microbial agents on the rhizosphere soil layer of tomatoes, which can evaluate the practical application effects of different carrier agents in agricultural production after 2 years of continuous tomato cropping. Secondly, the beneficial bacteria

identified in this study can provide theoretical support for the preparation of subsequent microbial preparations. Finally, this study can provide preliminary screening comparisons for the application of microbial agents in tomato intercropping disaster reduction measures in similar environments.

## MATERIALS AND METHODS

### Test materials and test site

This experiment was conducted between January 19, 2023 and February 17, 2023 at Xinsheng Seed Industry, Pingyuan County, Dezhou City, and from February 19, 2023 to September 2023 at the greenhouse of Dazhuang Village, Encheng Town, Pingyuan County, Dezhou City (120.4114°E, 30.4406°N), with 'Chun Li' tomato varieties as the test crop, and the previous crop in the test site was tomato. The basic physicochemical properties of the soil were organic matter (OM) 13.28 g·kg$^{-1}$, total nitrogen (TN) 1.12 g·kg$^{-1}$, effective phosphorus 125.55 mg·kg$^{-1}$, and fast-acting potassium 116.37 mg·kg$^{-1}$.

Microbial agents for testing: FQ_T1: *Bacillus subtilis* (Biovox Razen, wettable powder, effective live bacterial count was $1.00 \times 10^9$ CFU·mL$^{-1}$), FQ_T2: *Trichoderma harzianum* (Biovox Rootshield, wettable powder, effective live bacterial count was $3.00 \times 10^7$ CFU·mL$^{-1}$), FQ_T3: *Bacillus amyloliquefaciens* (Shanxi Xiannong Biotechnology Co., Xi'an, China; wettable powder, the effective number of live bacteria was $1.00 \times 10^9$ CFU·mL$^{-1}$), FQ_T4: *Verticillium chlamydosporium* (Yunnan Microstructure Source, micronutrient, the effective number of live bacteria was $2.50 \times 10^8$ CFU·mL$^{-1}$), FQ_T5: *Paecilomyces lilacinus* (Chongqing micro-nuclear biological purple warrior, wettable powder, the effective number of live bacteria was $1.00 \times 10^8$ CFU·mL$^{-1}$). Data were collected as previously described in *Wei et al. (2024)*. Five microbial fungicides were purchased from the biological plant protection station of the Guonong Agricultural Resources and Beihai Qunlin biological factory store. The test was conducted in a plastic greenhouse with double-layer thermal insulation cultivation facilities. The greenhouse was set up with a control (FQ_CK) and five microbial agent treatments (FQ_T1, FQ_T2, FQ_T3, FQ_T4, FQ_T5) planted on March 20, April 20, and May 19 *via* inter-root furrow application of five agents (dosage of 150 times the liquid, each time 100 mL per plant); control conventional management, no agents. Each experimental plot covered an area of 50 m$^2$, and experiments were repeated three times, randomly arranged, and the field management measures were the same as those of local high-yield greenhouses (*Wei et al., 2024*).

### Soil sample collection

Soil samples were collected on June 27th, 2023.

First, we removed the 0–3 cm of soil from the surface layer, and then took out the plants and surrounding soil to keep the root system intact. Second, we fully shook off the soil around the root system, gently brushed the soil close to the root system with a sterile brush to remove foreign matters, then randomly selected the rhizosphere soil of three tomato plants, mixed it and put into sterile plastic bags, which was one soil sample. The soil samples were divided into two parts after passing a 2 mm mesh sieve: one part was stored in an ultra-low temperature freezer at −80 °C for DNA extraction (*Liu et al., 2024*), and the

other part was placed in a cool and dry place for natural air drying and the determination of soil physicochemical properties and enzyme activities.

## Determination of soil physicochemical properties and enzyme activities

After high temperature digestion, the OM content of soil was measured using the redox method. The acid standard solution was utilized to determine the comcentration of TN through the semi-trace Kjeldahl method. The alkali nitrogen concentration was measured through the alkali diffusion method using an acid standard solution; available phosphorus (AP) content was extracted by 0.5 mol·L$^{-1}$ NaHCO$_3$ and then determined using the molybdenum antimony antimonimony anti-colorimetric method (*Xu et al., 2001*), and then detected by TU-1810 UV-Vis Spectrophotometer (Beijing Pudian General Instrument Co., Ltd., Bejing, China); and quick-acting potassium content was detected by TU-1810 UV-Vis Spectrophotometer. The content of AK was detected by CH$_3$COONH$_4$ extraction and then followed by a BWB-1 flame spectrophotometer (BWB Technologies, Hambridge Ln, Newbury, UK) (*Wei et al., 2024*).

Soil sucrase was determined using the colorimetric method of 3,5-dinitrosalicylic acid, soil catalase was determined using the titrimetric method of potassium permanganate; and alkaline phosphatase was determined using the Sodium Benzene Phosphate colorimetric method. All steps were performed by the China Rice Research Institute according to the instructions of the corresponding enzyme activity kits provided by Suzhou Keming Biotechnology Co. Ltd. A multifunctional enzyme marker (TECAN-Spark 20M) was used for colorimetric determination of the samples in Jiangsu, China (*Wei et al., 2024*).

## Analysis of microbial diversity measurements of rhizosphere soils of tomatoes

Using second-generation high-throughput sequencing technology (Illumina, San Diego, CA, USA) to sequence the 16S rRNA gene of tomato soil microorganisms; According to E Z.N.A. ® The instructions of the soil kit (Omega Biotek Company, Norcross, GA, USA) were used to extract the total DNA. The DNA mass concentration and purity were detected by NanoDrop 2000, and the DNA extraction quality was detected by 1% agarose gel electrophoresis; Bacterial 16S rRNA was amplified by PCR using 338F (5′-ACTCTAC GGAGCAGGGAGGCGAGGGGGAG-3′) and 806R (5′-GGACTAHVGGTWTCAAT-3′) primers for the V3–V4 variable region.

Illumina MiSeq sequencing: use 2% agarose gel to recover the PCR product, use AxyPrep DNA Gel Extraction Kit (American Axygen Biosciences Company, Union City, CA, USA) for purification, Tris HCl elution, and 2% agarose electrophoresis detection. Quantitative testing was performed using QuantiFluorTM ST (Promega, Madison, WI, USA). According to the standard operating procedures of Illumina MiSeq platform (Illumina, San Diego, CA, USA) (*Chen et al., 2018b*), the purified amplified fragments were used to construct a library 16S as a PE 300 library. All microorganisms in the soil were analyzed using QIIME software after double end sequencing was performed on Illumina's Miseq PE300 platform. Finally, annotation and analysis were performed using non
redundant protein sequence sets, non redundant protein amino acid sequence databases (NR), and carbohydrate active enzyme databases (CAZy) (Shanghai Meiji Biomedical Technology Co. Ltd., Shanghai, China). Species annotation were performed on the OTU sequences (*Bolyen et al., 2019*) and the community composition, alpha diversity, and relative abundance of the samples were counted at each taxonomic level, while heatmap, Mantel Test network analyses, and FAPROTAX function prediction analysis were performed within the platform on the correlation between the soil environmental factors and microbial communities to determine the effects of soil environmental factors on the composition of microbial communities (*Gu et al., 2024b*).

## Data processing and analysis of high-throughput sequencing results

Trimmomatic software was used for quality control of the sequenced raw sequences, and FLASH1.2.11 software was used for double-ended sequence splicing, and UPARSE7.1 software was used for OTU clustering of the sequences based on 97% similarity. Based on the OTU clustering results (*Francioli et al., 2020*), the Majorbio Cloud platform was used for further data analysis and information mining, including species community composition analysis, $\alpha$ diversity analysis (including Sobs index, Simpson index, Shannon-Wiener index, Chao1 richness, Abundance-based Coverage Estimator (ACE), coverage, and other indices), and performance of 16S gene function prediction analysis (COG database).

The relevant indicators are represented by the mean and standard error (SEM) of three biological replicates. The data were statistically analyzed using Excel 2007 and SPSS Statistics 26.0 software (IBM Inc., Armonk, NY, USA), and soil microbial diversity analysis and visual mapping were performed using Origin2019b and MajorBIO cloud platform (www.majorbio.com, Shanghai, China), and the OTUs were subjected to multiple sequence comparison. Analysis of variance and multiple comparisons ($p < 0.05$) were performed using one-way ANOVA and Duncan's Multiple Range Test (DMRT) method.

# RESULTS

## Effects of different microbial agents on soil physicochemical properties and enzyme activities of tomato

Different microbial agents have different degrees of effects on the physicochemical properties and enzyme activities of tomato rhizosphere soil.

The content of available potassium (AK) and organic matter (OM) in tomato rhizosphere soil treated with microbial agents showed significant differences relative compared to the control FQ_CK (Fig. 1): on average, the content of AK in tomato rhizosphere soil treated with five microbial agents was 50.56% higher than the control FQ_CK treatment, and the content of OM was 13.82% higher than with the control FQ_CK treatment. Except for the FQ_T2 treatment, there was a significant difference in TN content between tomato rhizosphere soil treated with microbial agents and the control FQ_CK treatment. When microbial agents were used to treat tomato rhizosphere soil, the TN content was on average 22.83% higher than the control FQ_CK treatment. Except for the FQ_T1 treatment, there was a significant difference in the AP content of tomato

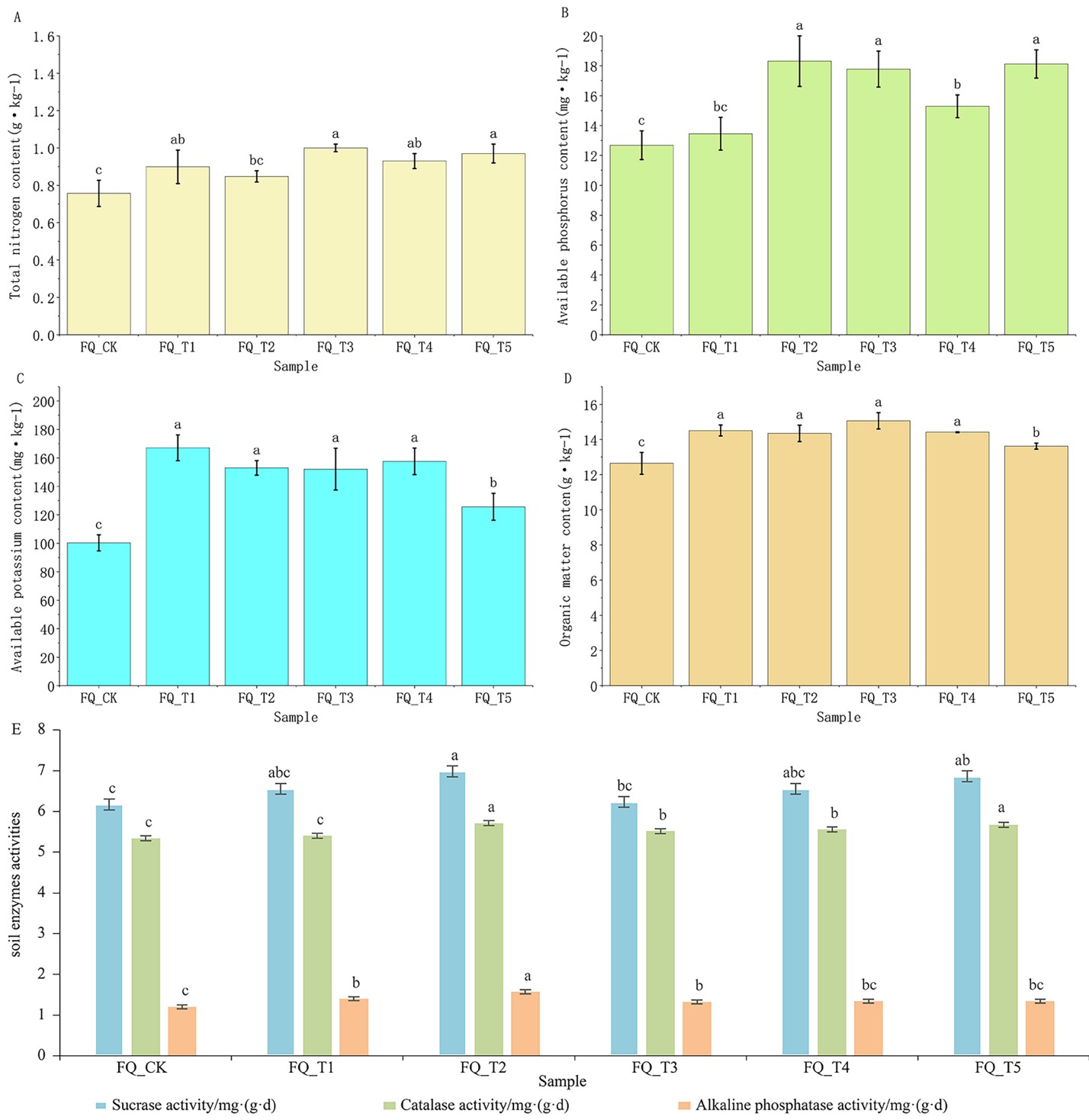

**Figure 1 Effects of different microbial agents on soil physicochemical properties and enzyme activities of tomato.** Note: Different lowercase letters above the bars indicate significant differences between treatments ($p < 0.05$).

rhizosphere soil after microbial agent treatment compared to the control FQ_CK treatment. The AP content of tomato rhizosphere soil treated with microbial agent was on average 30.81% higher than that of the control FQ_CK treatment.

Microbial agent treatment had a certain degree of impact on the enzyme activity of tomato rhizosphere soil. The combination of microbial agent treatments FQ_T2 and FQ_T5 showed a significant difference in the sucrose enzyme activity of tomato rhizosphere soil compared to the control FQ_CK treatment. The catalase activity (CA) in tomato rhizosphere soil treated with microbial agents FQ_T2, FQ_T3, FQ_T4, and FQ_T5 showed significant differences compared to the control FQ_CK treatment. The alkaline phosphatase activity (APA) in tomato rhizosphere soil treated with microbial agents FQ_T1, FQ_T2, and FQ_T3 was found to be significantly different from the control FQ_CK treatment. After treatment with microbial agents, the average activities of sucrose enzyme, catalase, and alkaline phosphatase in tomato rhizosphere soil were 7.59%, 4.39%, and 15.82% higher, respectively, than those in the control FQ_CK treatment. From this, it can be seen that although microbial agents have different effects on the physicochemical properties and enzyme activities of tomato rhizosphere soil, microbial agent treatment was overall beneficial at improving the physicochemical properties and enzyme activities of tomato rhizosphere soil in 2 years of continuous cropping (Fig. 1).

## Effects of different microbial agents on the microbial community in tomato rhizosphere soil

### Evaluation of sample sequencing depth and OTU cluster analysis

The dilution curve reached the plateau period, indicating that the sequencing data can cover the microbial community diversity and the dilution curve can also indirectly reflect species diversity richness. Sequencing of bacterial samples yielded a total of 248,652,103,952,463 bases (Fig. 2), with an average sequence length of 418 bases, and an OTU sequence similarity of 0.97. The dilution curve for the USEARCH11-uparse clustered OTU gradually plateaued after 25,000 sequences, suggesting an adequate supply of sequencing data and reliable outcomes, which was suitable for further analysis.

### Alpha diversity of soil bacteria treated with different microbial agents

Alpha diversity is the term used to describe the number and diversity of species in a local habitat or ecosystem, which is typically evaluated through Sobs, Simpson, Shannon, Chao1, and ACE indices. The Sobs and Chao1 indices reflect community richness, with higher index values indicating more species, while the Simpson index assesses community diversity, with a higher Simpson value indicates lower diversity. As shown in Table 1, the coverage of soil bacterial samples reached over 98.74%, confirming that the study's findings accurately represent the bacterial community diversity in the samples, and the sequencing results are deemed reliable. There were significant differences in ACE index, Chao1 index, and Sobs index between FQ_T1, FQ_T5 microbial agent treatment, and FQ_CK treatment. There was no significant difference in Simpson and Shannon index between the microbial agent treatment and control FQ_CK.

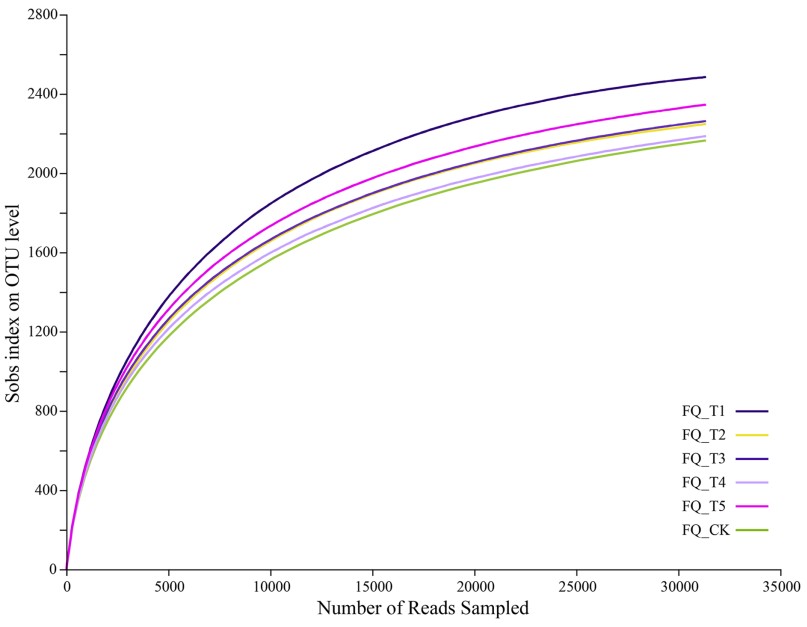

**Figure 2** Rarefaction curve.

**Table 1** Alpha diversity index of microbial community in rhizosphere soil of tomato.

| Sample | ACE index | Chao1 index | Shannon index | Simpson index | Sobs index | Coverage/% |
|---|---|---|---|---|---|---|
| FQ_CK | 2,449.07 ± 89.23c | 2,420.22 ± 63.12b | 6.49 ± 0.83a | 0.0036 ± 0.0024a | 2,164 ± 57.53c | 98.65 |
| FQ_T1 | 2,646.10 ± 16.55a | 2,578.20 ± 34.07a | 6.70 ± 0.26a | 0.0035 ± 0.0020a | 2,484 ± 79.61a | 99 |
| FQ_T2 | 2,475.28 ± 19.20c | 2,461.75 ± 14.95ab | 6.51 ± 0.82a | 0.0054 ± 0.0051a | 2,247 ± 6.94bc | 98.77 |
| FQ_T3 | 2,521.11 ± 33.80bc | 2,517.96 ± 39.22ab | 6.59 ± 0.22a | 0.0039 ± 0.0012a | 2,262 ± 88.43bc | 98.68 |
| FQ_T4 | 2,478.76 ± 60.89c | 2,475.56 ± 92.17ab | 6.42 ± 0.20a | 0.0078 ± 0.0038a | 2,186 ± 72.55c | 98.62 |
| FQ_T5 | 2,586.71 ± 41.64ab | 2,564.73 ± 79.42a | 6.68 ± 0.75a | 0.0031 ± 0.0011a | 2,345 ± 83.56b | 98.72 |

**Note:**
Different lowercase letters indicate significant differences between treatments ($p < 0.05$).

Overall, compared with the control FQ_CK, the application of microbial agents increased the community Sobs and Chao1 indices at different levels, indicating that the treatment of microbial agents increased the richness of bacterial communities and total number of species to a certain extent.

### Analysis of soil bacterial community structure after treatment with different microbial agents

The relative abundance of bacteria communities in the soil across all treatment groups changed significantly following the application of various microbial agents (Fig. 3).

Bacteria from 40 phyla, 130 classes, 309 orders, 490 families, and 877 genera were present in rhizosphere soil samples, resulting in a total of 1,598 species. As illustrated in Fig. 3, at the phylum level, 11 rhizosphere soil bacterial phylum exhibited relatively high abundances (relative abundance >1%), with *Proteobacteria* and *Firmicutes* as the predominant phyla, averaging 44.55% across all treatments. Compared to the FQ_CK

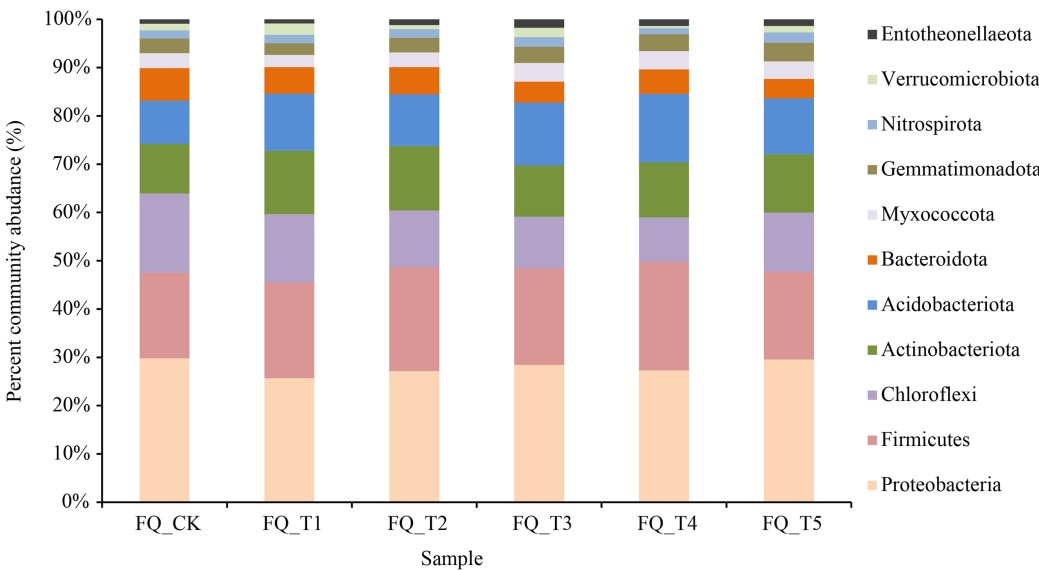

**Figure 3 Relative abundance under different treatments in phylum level.**

treatment, the five microbial treatments showed a decrease in the average relative abundance of *Proteobacteria* by 2.42%, with the FQ_T1 treatment experiencing the biggest drop at 14.61%. The FQ_CK treatment displayed a higher relative abundance of *Proteobacteria* at 28.03%. The FQ_CK treatment saw an increase in the average relative abundance of *Firmicutes* across the five microbial treatments increased by 2.22%, with the FQ_T4 treatment showing the greatest increase of 4.25%. The FQ_CK treatment had a larger relative abundance of *Chloroflexi* and *Bacteroideta* than any of the other five microbial treatments, with an average decrease of 4.69% and 1.79%, respectively. Meanwhile, the relative abundances of *Actinobacteria* and *Acidobacteria* increased by an average of 1.54% and 2.94%, respectively. Compared with the FQ_CK treatment, the relative abundance changes of *Myxococcata*, *Gemmatimonadota*, *Nitrospira*, *Verrucomicrobiota*, and *Entotheonellaeota* treated with five bacterial agents were slightly different, but the overall results showed an average increase of 0.21%, 0.16%, 0.05%, 0.02%, and 0.34%, respectively.

Figure S1 shows the dominant genus with relative abundance >1% at the genus level and ranking among the top 10. The 10 genera accounted for 27.03% on average among the treatments. Among them, *Bacillus* and *Paenisporosarcina* were the top two most dominant genera with relative abundance, and the unnamed genera accounted for 42.97% of the 10 genera. Compared to the FQ_CK treatment, the relative abundance of *Bacillus* increased by 52.52% on average. The relative abundance of *Paenisporosarcina* was 0.19 times higher than that of the treatment without application of fungicides for 2 years. Five microbial agent treatments had an average relative abundance of *Nitrospira* that was 2.91% higher than the FQ_CK treatment. In general, the application of microbial agents increased the relative abundance of *Bacillus*, *Paenisporosarcina*, and *Nitrospira* in tomato

rhizosphere soil in 2 years of continuous cropping and had a certain impact on the composition of bacterial community in tomato rhizosphere soil in 2 years of continuous cropping.

### Correlation between soil environmental factors and microbial community

To analyze the correlation between environmental factors and microbial community structure, Mantel test network heatmap analysis employs two commonly used matrices.

In Fig. 4, the absolute value of R (Mantel ′r) was used to plot the correlation between the bacterial community and environmental factors in tomato rhizosphere soil. The Mantel ′r of the bacterial community and AP and TN content was between 0.4–0.6, the absolute value of $P$ (Mantel ′p) $< 0.05$, and there was a positive correlation. There was a positive correlation between the bacterial community and the Mantel ′r of CA and AK content between 0.4 and 0.6, and Mantel ′p $\geq 0.05$. There was a positive correlation between bacterial community and OM, APA, Mantel ′r $< 0.4$, and Mantel ′p $\geq 0.05$. The bacterial community was negatively correlated with Mantel ′r $< 0.4$ and Mantel ′p $\geq 0.05$ of sucrase activity (SA). The seven environmental factors had distinct correlations, with the only significant positive correlation between CA and AP content.

### Analysis of microbial function prediction in tomato rhizosphere soil

FAPROTAX software maintains a functional classification database based on species information, which includes more than 80 functional classifications of carbon, nitrogen, phosphorus, sulfur, and other element cycles, as well as plant and animal pathogens, methane generation, and fermentation, covering more than 4,600 different prokaryotic species. The biochemical cycle processes of environmental samples can be predicted with good accuracy (*Louca, Parfrey & Doebeli, 2016*). Functional prediction of microorganisms is currently being carried out by many scholars using FAPROTAX (*Li et al., 2024*). This study also used FAPROTAX to analyze and predict the functions of bacteria in the rhizosphere soil of tomato treated with different microbial agents in the second year of continuous cropping and obtained 53 functional groups (Fig. 5). The analysis of functional bacteria with a relative abundance of >1% (average total bacterial proportion of 87.74%) showed that chemolithoautotrophic functional bacteria dominated in tomato rhizosphere soil (accounting for 24.98–33.39% of total bacteria), chemoheterotrophic and aerobic chemoheterotrophic bacteria are both included. The relative abundance of phototrophic bacteria was relatively low (0.45–1.91% of total bacteria), and the average relative abundance of phototrophic bacteria treated with the bacterial agent was 7.37% higher than that of the control FQ_CK treatment, which may be related to the promotion of the growth of phototrophic bacteria by the application of the bacterial agent.

The total relative abundance of functional microorganisms related to the nitrogen cycle (nitrate respiration, nitrate reduction, nitrogen respiration) was relatively high (averaging 8.02% of the total bacteria), and the average relative abundance of nitrogen cycle-related functional bacteria treated with microbial agents was 45.09% higher than that of the control FQ_CK treatment. This might mainly be due to the fact that the application of microbial agents offered a favorable growth environment for these bacteria. The relative
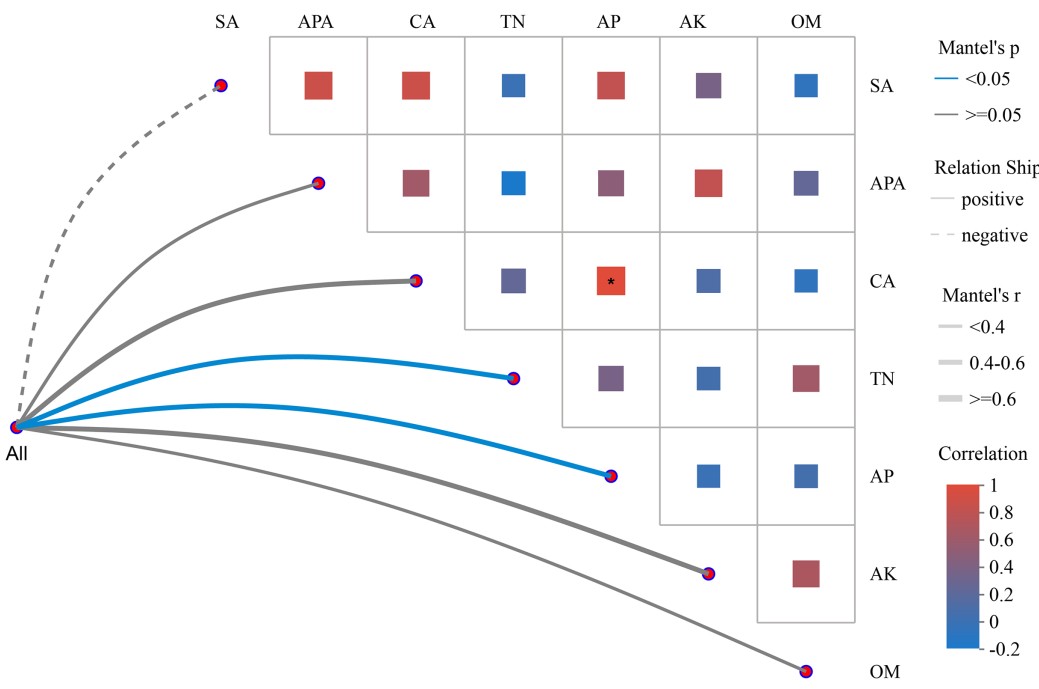

**Figure 4 Heatmap of Mantel test network analysis between tomato rhizosphere bacterial community and environmental factors.** Note: Mantel test heatmap: The lines in the graph represent the correlation between communities and environmental factors, while the heatmap represents the correlation between environmental factors. Line thickness: The correlation between community and environmental factors, plotted using absolute value of R (Mantel ′r). Relationship: Positive and Negative are positive and negative correlations between communities and environmental factors. In the heat map, different colors represent positive and negative correlations, color depth represents the magnitude of positive and negative correlations, and the asterisk in the color block represents significance, $*0.01 < p \leq 0.05$.

**Figure 5 Analysis of FAPROTAX functional prediction of tomato rhizosphere soil microorganisms.**

abundance of communities with fermentation functional genes (fermentation) ranged from 3.53% to 6.96%. The total relative abundance of functional microorganisms related to parasites (animal parasites, symbionts, and intracellular parasites) averaged 2.08% of the total bacteria. The total relative abundance of functional microorganisms related to decomposition (*xylanolysis*, *cellulolysis*, *chitinolysis*, *ureolysis*) ranged from 3.30% to 5.51%. The highest total relative abundance of decomposition-related functional bacteria was found in FQ_T1 with a 23.38% increase over the control FQ_CK treatment. Functional microorganisms related to human pathogenic bacteria (*e.g.*, human pathogens linked to pneumonia and other diseases, with an average total relative abundance accounting for 2.11% of the total bacteria) and those related to sulfur and manganese oxidation cycles (*e.g.*, dark thiosulfate oxidation, dark oxidation of sulfur compounds, and manganese oxidation, with an average total relative abundance accounting for 3.80% of the total bacteria) were also detected at the sampling sites.

## DISCUSSION

### Microbial agents are beneficial at improving the physicochemical properties and enzyme activity of tomato rhizosphere soil

The AK and OM contents in the rhizosphere soil of tomatoes treated with microbial agents showed a significant difference in comparison to those in the control FQ_CK treatment. The AK, OM, TN, AP contents and activities of sucrase, catalase, and alkaline phosphatase in the rhizosphere soil treated with microbial agents were all higher than those in the control FQ_CK group. Changes in soil microbial characteristics may have been the result of changes in soil chemical properties after 2 years of continuous cropping (*Wang et al., 2013*). In contrast, the application of microbial agents may have improved the physicochemical properties and enzyme activities in the rhizosphere soil, thereby improving the soil nutrient environment, increasing the abundance of beneficial microorganisms, and exerting microscopic regulatory effects on the ecosystem. This might be because some environmental variables have potential influences on the assembly of rhizosphere microbial communities and play a key role in the construction of rhizosphere microbial communities (*Chen et al., 2021*).

### Microbial agents can enhance the diversity of bacterial microorganisms in tomato rhizosphere soil

Ecosystem multifunctionality is influenced by Soil microbial diversity and network complexity, and soil microbial diversity plays a positive role in promoting soil multifunctionality (*Gong et al., 2024*). Compared to the control FQ_CK, the bacteriophage treatment led to an increase in the richness and total number of species of the bacterial community, to some extent. This is consistent with the research conclusion of *Wang et al. (2020)*, who conducted pot experiments on Hubei Haitang with *Bacillus* FKM10, which showed an increase in bacterial abundance and diversity and a change in the structure of the soil microbial community in the treatment compared to the control. The predominant phyla was *Proteobacteria* and *Firmicutes*. Compared with the FQ_CK, the relative

abundances of *Actinobacteriota*, *Acidobacteriota*, *Gemmatimonadota*, and *Nitrospirota* all increased. Previous studies have shown that *Actinobacteriota* can inhibit the activity of some pathogenic fungi, promote the activity of microorganisms in the soil that are beneficial to crops; and have a better salt tolerance to encourage the decomposition of plant residues, soil structure formation, seed germination, and root growth (*Henning et al., 2017*; *Sathya, Vijayabharathi & Gopalakrishnan, 2017*). The abundance of *Acidobacteriota* is highly correlated with soil nutrition, plays an important role in carbon and nitrogen metabolism (*Hou et al., 2018*), and is an important flora in promoting denitrification (*Chen et al., 2018a*). Previous studies have pointed out that the relative abundance of *Acidobacteriota* is closely related to soil pH, and the abundance of *Acidobacteriota* shows an increasing trend with the decrease of soil pH (*Lauber et al., 2009*). Changes in the relative abundance of *Acidobacteriota* can indicate different nutritional strategies of soil bacterial communities. *Nitrospirae* is greatly related to the oxidation and reduction of nitrate. Among them, *Nitrospira* as a nitrifying bacterium can oxidize nitrite to nitrate (*Feng, Wu & Xu, 2008*) and can play a key role in the decomposition of particulate OM (*García-López et al., 2019*). *Chloroflexi* has strong tolerance to barren and harsh external growth environments and is suitable for growth and reproduction in low-nutrient environments (*Zhou et al., 2015*; *Huang et al., 2019*; *Epelde et al., 2015*). *Acidobacteriota* has a significant inhibitory effect on soil organic matter mineralization (*Guo et al., 2017*). On the contrary, *Bacteroidetes* is the main participant in soil OM mineralization (*Guo, Gong & Guo, 2015*). This article found that the microbial agent treatments increased the abundance of *Acidobacteriota* and decreased the relative abundances of *Bacteroidetes* and *Chloroflexi*. The reason is that *Acidobacteriota* can achieve neutralization and regulation of alkaline soil through metabolic acid production (*Schmalenberger et al., 2013*) and is a beneficial bacterium in the plant salt resistance mechanism (*Xu et al., 2020*). Some *Acidobacterial* microorganisms, acting as plant growth-promoting rhizobacteria (PGPR), can produce plant growth-promoting hormones (auxin IAA) and have the ability to produce iron carriers, dissolve phosphate, and interact with plants, thereby promoting plant growth (*Kalam, Basu & Podile, 2022*). Therefore, the increase in the abundance of these beneficial phyla and the decrease in the abundance of harmful phyla after applying microbial agents can promote the development of soil bacterial communities in a useful direction and increase soil activity.

## An interactive relationship between the bacterial microbial community in tomato rhizosphere soil, soil physicochemical properties, and enzyme activity

The bacterial community was positively correlated with AP, TN, CA, AK, and OM content and APA. The bacterial community was negatively correlated with SA. The seven environmental factors had different correlation relationships among each other, and only CA and AP content had a significantly positive correlation relationship. A large number of studies have reported that different microbial agents can also change soil physicochemical properties and soil enzyme activities and promote crop growth. *Abdelraouf, Hussain & Naguib (2023)* greatly improved the soil enzyme activity in the rhizosphere of tomatoes

and reduced the infection of tomato *Fusarium* wilt by exogenous application of nanochitosan-encapsulated *Pseudomonas*.

## Microbial agents promote beneficial bacterial functions in the rhizosphere soil of tomato after 2 years of continuous cropping

Ecological network analysis, as an important means to clarify the interactions among microorganisms, holds great significance in exploring the assembly, construction, and stability of rhizosphere microorganisms during grassland degradation. The use of microbial agents increased the diversity of fungi and interaction and stability of species ecological networks, enhanced the ability of microorganisms to resist external environmental disturbances, and contributed to maintaining the stability and sustainability of farmland ecosystem functions (*Yu et al., 2022*). When using FAPROTAX to analyze and predict the functions of tomato rhizosphere soil bacteria after different microbial agent treatments in the second consecutive year, a total of 53 functional population groups were obtained. Functional bacteria with a relative abundance >1% (average total bacteria accounted for 87.74%) were selected for analysis. The results showed that chemoheterotrophic functional bacteria in the tomato rhizosphere soil were dominant (accounting for 24.98–33.39% of the total bacteria), including chemoheterotrophic and aerobic chemoheterotrophic bacteria. The average relative abundances of functional bacteria related to the nitrogen cycle and phototrophic bacteria were 45.09% and 7.37% higher, respectively than those of the control FQ_CK. The total relative abundances of functional bacteria related to fermentation and decomposition ranged from 7.28% to 11.43%.

At present, there are various types of single microbial fertilizers and compound microbial agents in production. The synergy of multiple microorganisms applied in the soil is also an important scientific issue. This study only conducted field experiments on five single microbial agents in the soil of 'Chunli' tomatoes. Whether it has the same effect on other varieties of tomatoes and more than two consecutive years of continuous cropping still needs further study. The interaction between soil microbial communities and environmental factors reflects the complex adaptation mechanisms and feedback regulation processes of ecosystems. The spatiotemporal heterogeneity of environmental factors continuously screens and shapes the structure of microbial communities, thereby regulating the functionality and system stability of ecosystems. However, this study found that the correlation between some environmental factors and microbial community structure did not reach a significant level ($p > 0.05$), This may imply: (1) the presence of other key but undetected environmental drivers; (2) Microbial interspecific interactions may have a regulatory effect on the response of environmental factors; (3) There may be synergistic or antagonistic effects between different environmental factors. Therefore, future research should focus on reconstructing and building a multidimensional environmental factor detection system, integrating multi-omics technologies such as metagenomics and metabolomics, and deeply analyzing the interaction network between microorganisms, environment, and ecological functions. On the one hand, it is necessary

to conduct in-depth research on the functional genomics of microorganisms, and understand their specific metabolic pathways and ecological functions, to better utilize them for continuous cropping obstacle reduction. On the other hand, further exploration should be conducted on the stability and sustainability of these species differences and the resulting ecosystem changes on a long-term scale, providing a theoretical basis for evaluating the long-term effects of continuous cropping obstacle reduction.

## CONCLUSION

This study showed that the application of different microbial agents had significant positive effects on soil physicochemical properties and enzyme activities. At the same time, the composition of the microbial community in the tomato rhizosphere soil changed with the application of microbial agents, and the increase in microbial diversity and the enrichment of beneficial bacteria in the tomato rhizosphere had certain effects on the interaction, stability of the microbial ecological network, and resistance to environmental changes. Therefore, in actual production, attention should be paid to maintaining the stability of the tomato rhizosphere soil microecology by applying microbial agents, which may have a certain alleviating effect on continuous cropping obstacles. The findings of this study offered a comprehensive demonstration of the relationship between microbial fertilizers and the diversity of the tomato root microbial community. This was valuable for further understanding the construction process and potential mechanisms of the tomato root microbial community when different microbial fertilizers were used. Additionally, it provided a theoretical foundation for the development of microbial fertilizers, particularly composite microbial fertilizers, for soil improvement. Ultimately, this research contributed to stabilizing the macroecology of root-soil.

## ACKNOWLEDGEMENTS

Thanks to everyone who helped with the experiment and the completion of the manuscript.

### Funding

This work was supported by the Youth Fund. The funders had no role in study design, data collection and analysis, decision to publish, or preparation of the manuscript.

### Grant Disclosures

The following grant information was disclosed by the authors:
Youth Fund.

### Competing Interests

The authors declare that they have no competing interests.

## Author Contributions

- Longxue Wei conceived and designed the experiments, analyzed the data, prepared figures and/or tables, authored or reviewed drafts of the article, and approved the final draft.
- Dongbo Zhao analyzed the data, prepared figures and/or tables, authored or reviewed drafts of the article, and approved the final draft.
- Lianghai Guo performed the experiments, authored or reviewed drafts of the article, and approved the final draft.
- Jianjun Guo performed the experiments, authored or reviewed drafts of the article, and approved the final draft.
- Jinying Zhu conceived and designed the experiments, authored or reviewed drafts of the article, and approved the final draft.
- Yanting Pei conceived and designed the experiments, authored or reviewed drafts of the article, and approved the final draft.
- Peiyan Guan performed the experiments, authored or reviewed drafts of the article, and approved the final draft.
- Zhihui Guo performed the experiments, authored or reviewed drafts of the article, and approved the final draft.
- Huini Cui performed the experiments, authored or reviewed drafts of the article, and approved the final draft.
- Jiansheng Gao performed the experiments, authored or reviewed drafts of the article, and approved the final draft.
- Yongjun Li analyzed the data, authored or reviewed drafts of the article, and approved the final draft.
- Liang Zhang analyzed the data, authored or reviewed drafts of the article, and approved the final draft.
- Fuyan Wang analyzed the data, authored or reviewed drafts of the article, and approved the final draft.
- Peng Liu conceived and designed the experiments, authored or reviewed drafts of the article, and approved the final draft.

## Data Availability

Data is available at PRNJA1191903.

The sequences are available at NCBI biosample: SAMN45096129, SAMN45096130, SAMN45096131, SAMN45096132, SAMN45096133, SAMN45096134.

## Supplemental Information

Supplemental information for this article can be found online at http://dx.doi.org/10.7717/peerj.19564#supplemental-information.

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
