# Peer review of "Interactive effects between soil properties and bacterial communities in tomato rhizosphere under the application of microbial agents"

_PeerJ, doi:10.7717/peerj.19564_

## Round 0.1 · original submission · Major Revisions

- Please improve the introduction by clearly identifying the gaps in existing research and explaining how this work addresses or bridges those gaps.

- The methodology is not clearly explained. Please revise and provide more details.
- Strengthen the discussion section by ensuring that it is well-supported by the results.
- Please check the comments in the attached file raised by the Reviewers.

Reviewer 1 ·

Basic reporting

.

Experimental design

.

Validity of the findings

.

Additional comments

Interactive Effects Between Soil Properties and Bacterial Communities in Tomato Rhizosphere Under the Application of Microbial Agents
In this by Liu et al., authors provide a valuable insight into the role of microbial agents in improving soil health and crop productivity under continuous cropping conditions. It addresses the challenges of continuous cropping in tomato cultivation, offering insights into sustainable agricultural practices. Their integration of 16S rRNA and high-throughput sequencing techniques and detailed physiochemical and enzymatic analysis provides a robust analysis of soil microbial communities. Despite these merits however, I think improvements in clarity, methodological transparency and a broader focus on depth of discussion would enhance its practical impact, scientific rigor and value for publication.
For the starters, the introduction is extremely heavy, overly detailed and thus challenging to identify research gap. Therefore, I’d focus on the knowledge gap. I’d highlight why this study fills an important gap in knowledge. There’s a sudden change in the intro also, I’d keep it concise to make it flow.
Line 69, name of the bacteria , fungi should be italicised.
What is ITS high throughput sequencing techniques?
Check line 134?
Check line 166? Seems extremely carelessly written. Same with line 318.
Figures lack detailed clear annotations as it should allow for standalone interpretation. Please improve the figure legend or mention it clearly in the main text.
I’d use Graphpad prism for figure 1 and 3 where you’d be able to show technical/biological repeats and error bars etc. Clearly state the statistical significance.
The rationale for selecting the five microbial agents is not clearly stated. Were these agents chosen based on previous studies or preliminary experiments? Details on the standardization of microbial agent applications such as preparation methods, quality control are sparse and should be elaborated.
Manuscript does not provide sufficient details on how statistical tests were performed.
The role of specific microbial taxa- Bacillus, P. nitrospira in promoting soil health is mentioned but not thoroughly explored.
The manuscript briefly acknowledges the limitations of using single microbial agents but does not delve into how this affects the generalizability of the findings.
There is limited discussion on the scalability and economic feasibility of applying microbial agents in large-scale agricultural settings.
Please explore the mechanisms by which microbial agents enhance soil properties and microbial diversity. Please provide more details on the experimental setup, such as how the microbial agent treatments were prepared and applied.
I’d include a justification for the selection of microbial agents and discuss any preliminary tests conducted.
I’d also discuss how the findings can be translated into actionable strategies for farmers, maybe? Please address potential barriers to adopting microbial agents in large-scale agriculture, including cost and compatibility with existing farming practices.
Referencing is not clear, and I’d follow a clear acceptable referencing style. Check each reference individually for doi, volume etc information.
There are minor grammatical errors, awkward and repetitive phrasing. I would rephrased those for the conciseness.
Keep up the good work and best of luck,
Cheers!

·

Basic reporting

Kindly, check the attached document

Experimental design

Kindly check the attached document

Validity of the findings

Kindly check the attached document

Additional comments

Kindly check the attached document

---

## Round 0.2 · Major Revisions

- Please provide a clear and detailed methodology.

- Kindly respond thoroughly to Reviewer 2. There are multiple questions raised by Reviewer 2 in the attached file. Please address all points carefully and relate them to your work.

Reviewer 1 ·

Basic reporting

I think the manuscript has been significantly improved. I appreciate the authors taking this critical criticism in positive spirit of science. This manuscript maybe thus suitable for the publication.
Good luck,
Cheers!

Experimental design

I think the manuscript has been significantly improved. I appreciate the authors taking this critical criticism in positive spirit of science. This manuscript maybe thus suitable for the publication.
Good luck,
Cheers!

Validity of the findings

I think the manuscript has been significantly improved. I appreciate the authors taking this critical criticism in positive spirit of science. This manuscript maybe thus suitable for the publication.
Good luck,
Cheers!

Additional comments

I think the manuscript has been significantly improved. I appreciate the authors taking this critical criticism in positive spirit of science. This manuscript maybe thus suitable for the publication.
Good luck,
Cheers!

·

Basic reporting

More revision is required on the mmanuscript

Experimental design

The methodology is not sufficient enough

Validity of the findings

The author need to check again comparing with the employed methods

---

## Round 0.3 · Minor Revisions

L74: The authors stated that both 16S rRNA and ITS were used for high-throughput sequencing. However, ITS was not mentioned in the Materials and Methods or the Results sections. Additionally, the authors indicated that only bacterial communities were analyzed.


There are some mistake in English usage. Please carefully check to improve the English quality.

---

## Round 0.4 · accepted · Accept

The manuscript has much improved and can be accepted for publication.